# Latent Class Analysis of Polysubstance Use and Sexual Risk Behaviors among Men Who Have Sex with Men Attending Sexual Health Clinics in Mexico City

**DOI:** 10.3390/ijerph19148847

**Published:** 2022-07-21

**Authors:** Rosibel Rodríguez-Bolaños, Ricardo Baruch-Dominguez, Edna Arillo-Santillán, Elsa Yunes-Díaz, Leonor Rivera-Rivera, Lizeth Cruz-Jiménez, James F. Thrasher, Alan G. Nyitray, Eduardo Lazcano-Ponce

**Affiliations:** 1Tobacco Research Department, Population Health Research Center, Mexican National Institute of Public Health, Cuernavaca 62100, Mexico; thrasher@mailbox.sc.edu; 2Inspira Cambio, Mexico City 06470, Mexico; ricardo.baruch@gmail.com; 3Cardiovascular Diseases, Diabetes and Cancer Research Department, Population Health Research Center, Mexican National Institute of Public Health, Cuernavaca 62100, Mexico; eyunes@insp.mx; 4Reproductive Health Department, Population Health Research Center, Mexican National Institute of Public Health, Cuernavaca 62100, Mexico; lrivera@insp.mx; 5Evaluation and Survey Research Center, Mexican National Institute of Public Health, Cuernavaca 62100, Mexico; lizethcruzj@gmail.com; 6Department of Health Promotion, Education & Behavior, Arnold School of Public Health, University of South Carolina, Columbia, SC 29208, USA; 7Center for AIDS Intervention Research, Medical College of Wisconsin, Milwaukee, WI 53226, USA; anyitray@mcw.edu; 8Clinical Cancer Center, Medical College of Wisconsin, Milwaukee, WI 53226, USA; 9Mexican National Institute of Public Health, Cuernavaca 62100, Mexico; elazcano@insp.mx

**Keywords:** men who have sex with men, substance use, risk behavior, tobacco use, alcohol use

## Abstract

Men who have sex with men (MSM) are more likely to use drugs and other substances compared to their heterosexual peers. No studies have evaluated patterns of substance use among MSM adults in Mexico. We used latent class analysis (LCA) to identify MSM subgroups with specific substance use patterns and their associations with sexual behaviors. Methods: Data from 1850 adult MSM were collected at HIV clinics in Mexico City between September 2018 and December 2019. The structural equation modeling approach was used to estimate a LC model to identify patterns of substance use by self-report of substance use (i.e., cigarette smoking, alcohol, and drugs). To evaluate LC membership, we included HIV status, condomless anal sex (CAS), and serosorting, while controlling for demographic variables. Results: 30.3% were under the age of 22. Alcohol use in last 30 days (76.2%), binge drinking (29.2%), marijuana (29.4%), sex-drugs (23.9%), stimulants (13.7%), and depressants (6.3%). MSM reported engaging in CAS (55.9%) and serosorting (13.5%) behaviors, and 40% reported being HIV positive. LCA indicated three general categories of MSM substance users: Class 1 (49.0%), Class 2 (29.8%), and Class 3 (20.4%). Members of Class 3 were younger: 23–28 age years (aOR = 1.86) and 29–33 age years (aOR = 1.86), more educated: completed graduate studies (aOR = 1.60), had a high probability of polysubstance use and were more likely to engage in CAS and serosorting. Conclusions: Attempts to detect alcohol and problematic use of substances are needed for MSM followed by culturally competent approaches that address alcohol and drug use disorders.

## 1. Introduction

The risk of acquiring human immunodeficiency virus (HIV) is, globally, 26 times higher among gay men and other men who have sex with men (MSM) [1]. The prevalence of HIV among MSM ranges between 7% and 20% [2], reaching 17.3% in Mexico [3]. While substance use is relatively high among MSM adults [3,4], no studies have evaluated patterns of substance use and risk behaviors among MSM adults in Mexico. The current study uses latent class analysis (LCA) to identify groups of MSM in Mexico who exhibit relatively distinct patterns of substance use.

According to studies from other countries, MSM, compared to other males [5], are more likely to use substances, including cigarette smoking [6,7], binge drinking alcohol [7,8], and marijuana use [7,9,10]. Furthermore, MSM who use drugs have a higher prevalence of using drugs during sex in comparison with the general population [11,12,13,14], including methamphetamine (crystal meth), erectile dysfunction drugs (EDDs), and volatile nitrates (poppers). The use of these substances often co-occurs with anal sex without condoms or sex with multiple partners [15]. In particular, the use of crystal meth by MSM has been linked to a rise in the practice of condomless anal sex (CAS) behaviors [16,17,18] and serosorting behavior (the practice of only having sexual relations with partners of the same serostatus) [19]. 

Mexico has a population of 123 million [20]; 3.2% of the population aged 18 and older identify as non-heterosexual (lesbian, gay, or bisexual) [21]. The LGBT–Mexican survey [4] reported that among gay and bisexual individuals who responded to an online questionnaire, over 70% of used marijuana, 88% used alcohol, 45% used poppers, 25% used cocaine and crystal meth, 11% used LSD, and less than 10% used other drugs (gamma hidroxibutirato (GHB), ketamine) in the last year. Thirty-eight percent reported engaging in CAS behaviors while being severely intoxicated by drugs. Among men living with HIV, 89% reported alcohol use, and 72% reported drug use, including marijuana, ketamine, poppers, and crystal. Other than through this convenience sample recruited through social media, substance use among gay and bisexual men has not been studied; this study only evaluated the prevalence of using each substance, evaluated one at a time (not poly-use).

Latent class analysis (LCA) can be used to evaluate associations between person-level interrelated factors to identify a small set of qualitatively unique classes of people. Estimated proportions of latent classes can provide information on the prevalence of how certain behaviors are grouped. For example, because LCA identifies unobservable groups with a population or subpopulation, interventions can be tailored to target the subgroups that are most in need and who may benefit the most from the intervention. The application of the LCA is to determine unique subgroups of MSM who are at high risk and to refine public health messages about the harms and consequences of substance use. Previous studies in high-income countries (HICs) have used LCA to classify MSM based on substance use patterns. Low-risk subgroups include no drugs, tobacco, alcohol, or marijuana use [7,22,23]. Other subgroups reporting polydrug use (e.g., cocaine, inhalants, mushrooms) and drug use during sex (e.g., crystal meth, EDDs, poppers) [24,25,26] are at higher risk of engaging in CAS and of getting HIV.

There is limited information regarding the relationship between polysubstance use and sexual risk among MSM in Mexico. Studies in France have proposed a conceptual model to suggest that proximal factors within sexual relationships, such as age, HIV status, substance use, and partner type, could affect sexual risk behavior. This conceptual model hypothesized how the interplay between intercourse dynamics, position preferences, and substance use impact subsequent HIV/STI risks among MSM. For example, the odds of being classified as majority top/serosorters increased with increased age [27].

The objective of this study was to identify MSM subgroups (or latent classes) with specific substance use patterns and their associations with behaviors that may make them more vulnerable to contracting HIV (i.e., CAS, serosorting) and sociodemographic characteristics.

## 2. Materials and Methods

### 2.1. Data Source

We analyzed data from a cohort clinic-based study from two clinics in Mexico City that offer prevention services and specialized medical care for sexual minority people: “Prevention and control of neoplasms associated with HPV in high-risk groups in Mexico City: The Condesa Study” [28]. The non-probability sample (i.e., for convenience) was collected between May 2018 and December 2019; clinic users who met the following criteria were invited to participate: (a) between 18 and 60 years old, (b) designated male at birth, (c) having had a previous sexual encounter with a man, or (d) identify as gay or bisexual. A total of 2027 participants who were HPV-negative or HPV untested were recruited, including 177 transgender women and 1850 MSM. The present study includes only data from 1850 MSM who responded to the first survey to ensure that prior survey participation did not influence responses. In other words, we did not include data from subsequent surveys.

All data were collected through a computer-administered questionnaire. The participants provided informed consent before beginning the survey. Approval was obtained by the Institutional Review Board and Ethics Committee of the National Institute of Public Health of Mexico.

### 2.2. Measures

#### 2.2.1. Demographic Characteristics

MSM were defined based on self-declared sexual behavior (having had sex with a man). The participants reported their age, which was recoded into quartiles by distribution (i.e., 18–22, 23–28, 29–33, and 34+ years old). Marital status was coded in two categories: single vs. married or common-law. Educational attainment was classified into three categories based on the number of years of education since starting primary school: high school graduates (12 years or less of education), 4-year College graduates (13–16 years), and completed postgraduate studies (17 or more years). Self-reported HIV status was also classified into three categories (unknown, negative, or positive).

#### 2.2.2. Cigarette Smoking

Current smoking frequency was assessed by asking: “During the past 30 days, have you smoked cigarettes every day, have you smoked some days, or do you currently not smoke?” Responses were coded into three categories: nonsmoker (including former smokers?), nondaily smoker (some days), and daily smoker (every day) [29].

#### 2.2.3. Alcohol Use and Binge Drinking

Alcohol use was assessed based on the following question: “In the last 30 days, have you had a drink of alcohol (beer, pulque, wine, brandy, whisky, rum, tequila, coolers, NewMix, Cubaraima, etc.)?” (yes vs. no). Binge drinking was assessed based on the following question: “When you drink alcoholic beverages such as beer, spirits, coolers, etc., generally, how many glasses do you have on each occasion?” If they reported 5 or more drinks, it was considered binge drinking (yes vs. no) [29].

#### 2.2.4. Drug Use

Drug use in the last previous months was evaluated by asking “What drugs have you used in the last 3 months?” for which participants could indicate all options that applied: marijuana, cocaine, traumazol, benzodiazepine, GHB, mushrooms, ketamine, inhalants, opioids, crystal meth, erectile dysfunction drugs (EDDs), poppers, and other drugs. The participants’ responses were categorized into four dichotomous variables if they had ever used at least one drug in each group: (1) marijuana was evaluated separately in our analysis due to its high prevalence, (2) sex-drugs (methamphetamine (crystal meth), traumazol, EDDs, and volatile nitrates (poppers)), (3) stimulants (cocaine, crack, ketamine, amphetamines), and 4) depressants (tranquilizer: benzodiazepines, rivotril, Ghb, opioids, inhalable).

#### 2.2.5. Sexual Behavior

The participants were asked whether, in the previous 3 months, they had condomless anal sex: Have you practiced “bareback” (CAS by choice)? (yes vs. no). With regard to whether the participants had engaged in “serosorting” [30,31], it was evaluated based on the following question: Have you ever engaged in “serosorting” (unprotected sex with an HIV-positive person)? (yes vs. no).

### 2.3. Data Analysis

We performed a descriptive analysis of the participants’ sociodemographic characteristics, sexual behaviors, cigarette smoking, alcohol use, and drugs use. The structural equation modeling approach was used to estimate a latent class model to identify patterns of substance use by self-reporting of substance use (e.g., cigarette smoking, alcohol, and drugs). This method helped us detect the smallest number of relatively homogenous groups regarding substance use patterns in the analytic sample and generated estimates on the sizes of the groups and the joint probabilities of the conditioned response vectors for the latent class of membership [32]. To evaluate membership in each class, in the modeling we included sociodemographic characteristics (age in quartiles, marital status, and educational attainment), HIV status, and sexual behaviors (CAS and, serosorting), and estimated a significance level of *p* < 0.05 adjusted odds ratio (aOR). Firstly, through an iterative modeling process, we estimated a single latent class, which was considered the baseline model, and we specified an additional class each time to produce 2 to 4 latent class models. Then, to define the best number of latent classes in our model, we evaluated the relative fit of the models using the Bayesian information criterion (BIC) [33] (Table 1). This information criterion helped us to correctly identify the number of classes with sensitivity to small sample sizes, regardless of the model type. Finally, we reported the model with three classes. All analyses were performed using STATA v.16 (StataCorp LP, College Station, TX, USA) [34].

## 3. Results

### 3.1. Descriptive Statistics of the Study Population

The characteristics of the participants across analyses are summarized in Table 2. The median age was 29 years (Q1, Q3: 18, 34); 30.3% of the sample was under the age of 22 and 80.0% were single. Among the participants, 90.1% had a high level of educational attainment with 13 years or more of education (postgraduate college). Almost half (40.6%) were men living with HIV, 55.9% engaged in CAS behaviors, and 13.5% had engaged in serosorting behavior. The most commonly reported substances used were alcohol (76.2%), including binge drinking (29.2%), followed by non-daily smoking (29.2% both), marijuana use (29.4%), sex-drugs (24.2%), stimulants (13.7%), and depressants (6.3%).

### 3.2. Latent Class Analysis

Three groups were identified by LCA in our sample (Figure 1).

Class 1. The first group was identified as low substance use, accounting for 49.0% of participants, and characterized by current alcohol use (59.9%).

Class 2. A total of 31.4% of all respondents were included in this group. This group had a relatively high probability of alcohol use (96.6%) and binge drinking (54.5%) in the previous 30 days and had the highest likelihood of non-daily cigarette smoking (51.9%).

Class 3. A total of 19.6% of the respondents were classified into this group. This group was identified as the polysubstance group and was characterized by relatively high use of all substances, particularly alcohol (89.6%), marijuana (69.0%), sex-drugs (87.2%), stimulants (45.9%), and depressants (24.9%) when compared to the other groups.

Table 3 shows the results of the multinomial logistic regression analyses. When predicting membership in the Class 2 group, only age was statistically significant, as those who were older (adjusted odds ratio (aOR) = 0.48, 95% CI: 0.29–0.80) were less likely to be in that group than in Class 1. Membership in the Class 3 group was significantly more likely among MSM if they were younger: 23–28 age years (aOR = 1.86, 95% CI: 1.16–3.00) and 29–33 age years (aOR = 1.86, 95% CI: 1.14–3.03), more educated: completed postgraduate studies (aOR = 1.60, 95% CI: 1.05–2.43), or engaged in either CAS behavior (aOR = 2.92, 95% CI: 1.30–3.12) or serosorting (aOR = 2.02, 95% CI: 2.02–4.22).

## 4. Discussion

Our results show that among the MSM who attended sexual health clinics in the study period, and who were, on average, 29 years old, 40% reported being HIV positive and engaging in CAS risk behaviors. Almost a quarter (24.2%) of these men reported having used one or more sex drugs in the past three months, 76.2% used alcohol, and more than a quarter (29.2%) reported binge drinking. The MSM in this sample were classified into three general LCA categories of substance users: Class 1 principally identified alcohol users in the previous 30 days; Class 2 identified non-daily smokers, alcohol users, and binge drinkers; and Class 3 identified polysubstance users (smoking, alcohol, marijuana, and drugs). Our results demonstrate that members of the Class 3 group were characterized by a high probability of using alcohol and sex-drugs: crystal meth, traumazol, EDDs, and poppers. In the adjusted models, being a member of the Class 3 group was associated with having greater sexual risk behaviors (CAS behaviors) and serosorting, being younger, and having a high level of educational attainment.

Prior studies have demonstrated that some MSM are motivated to drink to enhance sex, which is also associated with greater alcohol use [7,35,36]. All LCA profiles had markedly higher likelihoods of alcohol use in the last month. However, in Class 2, more than 55% engaged in binge drinking. It is known that alcohol consumption is associated with AIDS-related deaths [37]; research has shown that alcohol use, especially binge drinking, is associated with an increase in CAS behaviors [38,39]. We found no association in CAS behaviors and serosorting between Classes 1 and 2, possibly because they are less likely to use sex-drugs.

Our results show a high use of drugs among young MSM who attend sexual health clinics, which is consistent with other existing studies [13,22,26,40]. Class 3 reported using marijuana and a wide range of drugs, including drugs during sex, i.e., crystal meth, EDDs, traumazol, and poppers, as well as other drugs: cocaine, benzodiazepine, GHB, mushrooms, ketamine, inhalants, and opioids. This is consistent with studies that have used LCA, where membership in a higher-risk subclass is due to a high proportion of drug use. For example, in Chan et al., Class 3 had the highest number of MSM who reported consuming alcohol or drugs during sex [40]. McCarty-Caplan et al. found that four latent classes, defined as ‘low drug use’, ’moderate drug use’, ’sex drug use’, and ‘polydrug use’, were classes that showed higher probabilities of using all other drugs assessed [13]. Lim et al. found three latent classes: ‘negligible substance use’, whose members reported having no substance use or using any substance in moderation; ‘soft substance use’, whose members used poppers, ecstasy, and alcohol before sex; and ‘amphetamine-type stimulants’ (ATS), whose members used stimulants (crystal meth, ecstasy), EDDs, and recreational drugs before having sex. The types of drugs used by MSM in these last two groups were markedly different. The men in the soft substance use group mostly used poppers and ecstasy, while those in the ATS group used crystal meth, ecstasy, ketamine, EDDs, and recreational drugs before sex [26]. Achterbergh et al. identified five latent classes of users: ‘no substance users’, ‘alcohol users’, ‘nitrite and EDD users’, ‘polydrug users’, and ‘chem-users’. The members of the last two classes were those with the highest consumption of drugs, especially sex-drugs: GHB/GBL, crystal meth, ketamine, and mephedrone [22].

We found subgroups of MSM who reported tobacco use. Card et al. evaluated substance use contextualized by sociocultural experiences, and they identified six latent classes: ‘assorted drug use’, ‘club drug use’ (e.g., ecstasy, mushrooms, LSD), ‘illicit drug use’ (e.g., heroin, codeine, oxycodone, crack), ‘sex drug use’ (e.g., GHB, poppers, EDDs, crystalline methamphetamine), ‘conventional drug use’ (e.g., tobacco, alcohol, and marijuana), and ‘limited drug use’. Card et al. found that conventional drug use was 100% for alcohol use, 80% for marijuana use, and 43% for tobacco use. Compared to our data, the prevalence of use of these substances was lower for alcohol (76%) and marijuana (29%), but the same for tobacco (non-daily smokers—29% and daily smokers—13%). Perhaps because of social norms, regulations, and law enforcement related to the use of these substances [41]. In another study, Newcomb et al. [6] conducted research on young MSM and identified three classes: ‘alcohol and marijuana users’, ‘polysubstance users’, and ‘low marijuana users’. They found a low (12.6%) prevalence of daily smokers. Although it is not possible to make a comparison due to the measurement of variables and age, the proportion of smokers in our sample was low compared to the proportion of current smokers (16.8%), only in Class 3 was it higher than the proportion of current smokers in Mexico [42].

Thus, in general, the typologies of substance use by MSM in Mexico appeared similar to studies conducted in Malaysia [26], France [27], and Chicago, USA [13]. MSM in this study reported a high probability of using drugs, suggesting that some MSM in Mexico use drugs for sexual encounters. With the advance in antiretroviral treatments for HIV infection, the perception of risk has changed and has produced an increase in sexual risk behaviors [43,44]. In our study, membership in Class 3 was associated with practices such as CAS behaviors and serosorting. A longitudinal cohort study between HIV-positive and HIV-negative MSM who were frequent users of alcohol and crystal meth showed that they had more episodes of condomless sex, thereby failing to follow through on their original intentions to engage in sexual intercourse with partners of the same serostatus [19]. In a study conducted in France [27], three latent classes of sexual positioning and serosorting profiles were identified: ‘majority top/serosorters’, ‘versatile/low partners’, and ‘majority bottom/some serosorters’. The majority top/serosorters group had the highest probability of condomless serosorting (72%), although the authors found no association with drug use before or during sex.

A review of the literature on CAS behaviors [45] showed that, in regard to such behaviors, multiple factors converged, including homonegativity, community norms, intimacy with a partner, and drug use. Likewise, HIV-positive serostatus might be a predictor of CAS sex. Halkitis reported that a large number (41.2%) of self-identified CAS participants in his sample were HIV positive [46]. Tobin et al. found three latent classes: ‘poly-users’, ‘alcohol/marijuana users’, and ‘low substance users’ [23]. The results showed that both the alcohol/marijuana and poly-user classes were significantly more likely to engage in sex under the influence of drugs and alcohol. In Mexico, a study found that MSM engaging in CAS behaviors was due to a change in risk perception, with access to antiretroviral therapy [47]. Different studies in Mexico have shown growing concern for addiction to crystal meth as a particular drug that may potentiate new cases of HIV among MSM due to its potential to enhance sexual desire and CAS with multiple partners [48,49].

Strategies to detect and address alcohol and substance use disorders should be considered based on a comprehensive approach that considers the needs of gay, bisexual, and other MSM, including their mental health. Risk reduction and harm reduction strategies can be particularly beneficial in preventing drug-related harm (e.g., overdose, addiction) and sexual health problems, including HIV and STIs [50]. In 2020, the brief guide for the implementation of the risk and harm reduction approach with drug users was published, in which a series of strategies was provided, such as sharing information on the effects of drugs and their possible combinations and information on establishments to access counseling or treatment, antiretroviral therapy, as well as condom distribution programs; interventions must also be carried out in nightclubs and festivals in order to reduce risks. These are some of the actions that could have positive results for MSM [50].

Governments and healthcare providers must implement programs that are sensitive to people with diverse sexual orientations and practices, without discrimination or stigmatizing views regarding drug use, preserving the right to health protection and non-discrimination, to contribute to the consolidation of a culture that respects human rights [51].

### Limitations

Our study is subject to limitations. Some of the participants did not provide information on some behavioral variables used in the LCA (e.g., tobacco products). The results of this study are based on patients presenting for care at a clinic, and such patients may differ from the general population. Although the results of this study may not be generalizable to other populations of MSM, the method for identifying subpopulations that may benefit from polydrug use interventions could be replicated in other populations. Our findings suggest that LCA helped identify a group of MSM at increased risk of vulnerability. Future replication of the analysis in larger independent samples would be useful in determining whether the same profiles reliably emerge. In addition, future studies should consider the existence of new substances on the market used by MSM, such as 2CB, choco-fungi, and synthetic mescaline. Future work should also consider using LCA to assess HPV infections and STIs to determine risk factors in a more rigorous way.

## 5. Conclusions

The use of alcohol and drugs among MSM directly affects sexual practices and, therefore, they may have a higher risk of contracting HIV than the general population. In particular, our LCA identified one group of MSM (Class 3) with a high probability of polysubstance use and greater engagement in CAS and serosorting. More research is needed to understand the dynamics of Class 3, given the increased risks that they experience due to their polydrug use, as seen in other countries where chemsex is popular among MSM.

This study shows the importance of assessing substance use in MSM to develop more tailored solutions for their sexual health and wellbeing, regardless of their HIV status.

## Figures and Tables

**Figure 1 ijerph-19-08847-f001:**
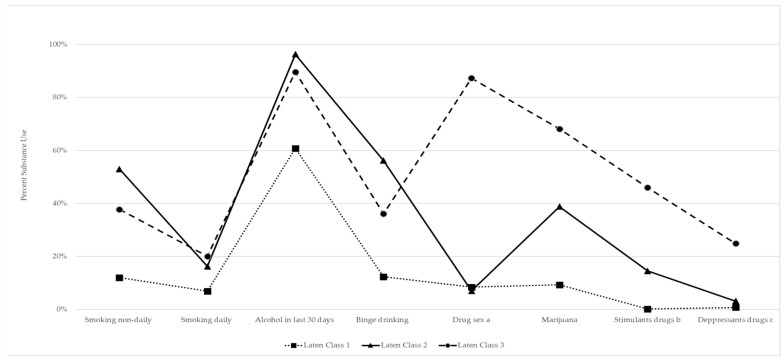
Patterns of smoking, alcohol, and drug use in MSM across three latent classes. Mexico City, 2018–2019. a. Includes methamphetamine (crystal meth), traumazol, erectile dysfunction drugs (EDDs), and volatile nitrates (poppers). b. Includes cocaine, crack, ketamine, amphetamines. c. Includes benzodiazepines, rivotril, Ghb, opioids, inhalable.

**Table 1 ijerph-19-08847-t001:** Model fit to identify an optimal number of latent classes.

Model	df	AIC	BIC
One class	8	13,543.42	13,587.21
Two classes	25	12,735.85	12,872.71
Three classes	42	12,569.68	**12,799.60**
Four classes	59	12,526.34	12,849.31

AIC = Akaike information criterion, BIC = Bayesian information criterion. Bolded = Indicates the lowest BIC value (i.e., the best number of classes).

**Table 2 ijerph-19-08847-t002:** Characteristic demographics, HIV status, sexual behavior, tobacco use, alcohol use, and drug use of 1850 MSM visiting HIV clinics in Mexico City, 2018–2019.

Characteristics	*n*	%
Age in Years	(*M* = 29.1, *SD* = 7.37)	(Range = 15–59)
18–22	561	30.3
23–28	446	24.1
29–33	392	21.2
34 or more	451	24.4
Marital status		
Single	1480	80.0
Married/common-law	366	19.8
Missing	4	0.2
Educational attainment		
High school graduates	545	29.9
Four-year college graduates	677	37.1
Completed postgraduate studies	604	33.1
HIV status		
Negative	1054	57.0
Positive	751	40.6
Unknown	45	2.4
Condomless anal sex (CAS)		
No	810	44.1
Yes	1028	55.9
Serosorting		
No	1592	86.5
Yes	248	13.5
Smoking pattern (last 30 days)		
Non-smoker	1068	57.7
Non-daily	541	29.2
Daily	234	12.6
Missing	7	0.4
Alcohol use (last 30 days)		
No	428	23.1
Yes	1410	76.2
Missing	12	0.6
Binge drinking (last 30 days)		
No	1245	67.3
Yes	541	29.2
Missing	64	3.5
Drug use (last 3 months) ^a^		
No	803	43.4
Yes	1021	55.2
Missing	26	1.4
Sex-drugs ^b^	434	23.9
Marijuana	544	29.4
Stimulants ^c^	249	13.7
Depressants ^d^	114	6.3

NOTE: M = mean; SD = standard deviation. ^a^ Based on multiple responses; therefore, percentages may add up to more than 100. ^b^ Includes methamphetamine (crystal meth), traumazol, erectile dysfunction drugs (EDDs), and volatile nitrates (poppers). ^c^ Includes cocaine, crack, ketamine, amphetamines. ^d^ Includes tranquilizers: benzodiazepines, rivotril, Ghb, opioids, inhalable.

**Table 3 ijerph-19-08847-t003:** Multinomial logistic regression predicting class membership relative to Class 1 of 1760 MSM visiting HIV clinics in Mexico City, 2018–2019.

Class 1–Reference	Class 2		Class 3	
	aOR	CI 95%	aOR	CI 95%
Age in Years				
18–22	1.00		1.00	
23–28	1.36	(0.90–2.05)	1.86 *	(1.16–3.00)
29–33	0.95	(0.59–1.52)	1.86 *	(1.14–3.03)
34 or more	0.48 *	(0.29–0.80)	1.54	(0.95–2.47)
Marital status				
Single	1.00		1.00	
Married/common-law	0.86	(0.56–1.29)	0.79	(0.53–1.16)
Educational attainment				
High school graduates	1.00		1.00	
Four-year college graduates	1.15	(0.77–1.71)	1.22	(0.80–1.86)
Completed postgraduate studies	0.67	(0.43–1.06)	1.60 *	(1.05–2.43)
HIV status				
Negative	1.00		1.00	
Positive	0.82	(0.5–1.15)	1.26	(0.90–1.75)
Condomless anal sex (CAS)				
No	1.00		1.00	
Yes	1.28	(0.89–1.84)	2.92 **	(1.30–3.12)
Serosorting				
No	1.00		1.00	
Yes	1.09	(0.62–1.89)	2.02 **	(2.02–4.22)

NOTE: aOR = adjusted odds ratio. CI = confidence interval. Model adjusted for all variables (age, marital status, educational attainment, HIV status, CAS and serosorting). * *p* < 0.05. ** *p* < 0.001.

## Data Availability

Not applicable.

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
