# Peer review of "Latent Class Analysis of Polysubstance Use and Sexual Risk Behaviors among Men Who Have Sex with Men Attending Sexual Health Clinics in Mexico City"

_ijerph, 2022, doi:10.3390/ijerph19148847_

Round 1
Reviewer 1 Report
This is a very important research study. It serves as a critical reminder of the effects of alcohol and drugs on sexual behavior among men having sex with men.
The literature review is very good. The sample size is excellent.
I am impressed with the level of detail regarding the different types of drugs and impacts on sexual behavior. That said, this is an article that will be appreciated by those with research interests in substance abuse and sexual behavior. I usually prefer a more simple presentation but see here the importance of providing significant detail.
The definition of latent class analysis (LCA) in lines 84 to 90 is especially strong.
I do not have expertise in statistical analysis and hope that another reviewer can be helpful to the authors.
The limitations section is very good. The authors make the important point that this work can help others to identify other populations that can benefit from polydrug use interventions.
The only suggestion I have is to expand the Conclusions section. After all of this hard work, the authors need to share their ideas regarding "actions to detect and address alcohol and substance use disorders... in gay, bisexual and other MSM." The authors also need to share their recommendations for risk reduction strategies.
Author Response
Thank you very much reviewing our manuscript. We have incorporated most of the suggestions made by the reviewers. Those changes are highlighted within the manuscript.

Reviewer 2 Report
Abstract
Line 33: The phrase “…a higher level of education” does not make sense: Please provide specific education status such as high school graduates, 4-year College graduates, or completed graduate studies, etc.
Method
What is the research design? Page 3, lines 107-111 states, “In this cross-sectional study, data were analyzed from “Prevention and control of neoplasms associated with HPV in high-risk groups in Mexico City: The Condesa Study”, which is a longitudinal, clinic-based study that recruited HPV-negative or HPV un-tested participants from two clinics in Mexico City. These clinics offer prevention services and 110 specialized medical care for sexual minorities people. The methodological details have been published elsewhere (Lazcano-Ponce et al., 2018). “ bold italics added by the reviewer. The claim that a cross-sectional study was designed out of a subset of data collected for a longitudinal study is not tenable. The authors need to provide a logical explanation of their method of data collection and their study design.
What is the research question? What is the knowledge gap being investigated?
How does this research contribute to filling that gap?
What is the sampling method used? How were the samples selected?
Results
The results section should include an adequate narrative of the findings presented in each table and figure, and stand-alone tables and figures do not have much to offer without an accompanying narrative of significant findings. The discussion should also focus on how the findings in this research compare with findings by other researchers for the same study area or similar research in other study areas.
Conclusion
There is no mention of patterns of polydrug use in MSM subgroups or latent classes in the conclusion section but the objective of the study was started, “…. to identify MSM subgroups (or latent classes) with specific substance use patterns and their associations with behaviors that may make them more vulnerable to contracting HIV….”
Reference
Alexis M. Roth, Richard A. Armenta, Karla D. Wagner, Scott C. Roesch, Ricky N. Bluthenthal, Jazmine Cuevas-Mota & Richard S. Garfein (2015) Patterns of Drug Use, Risky Behavior, and Health Status Among Persons Who Inject Drugs Living in San Diego, California: A Latent Class Analysis, Substance Use & Misuse, 50:2, 205-214, DOI: 10.3109/10826084.2014.962661
Author Response

(The authors gave the same response as above.)

Reviewer 3 Report
This study assessed the relationship between substance use patterns and sexual behaviors among MSM adults in Mexico using Latent Class Analysis. Overall, the findings from this paper are interesting since it may contribute to the poly substance use literature especially in MSM to prevent harmful sexual outcomes or diseases such as HIV and STIs, but many things need to think about and be improved.
1. For abstract, reduce the descriptive results (e.g., proportions) and save more space to report the main findings is necessary (e.g., ORs for age in Class 3, etc.).
2. For introduction, “While substance use is relatively high among MSM adults”, please add citations.
3. Suggest reframing the last sentences in first paragraph (“While substance use is relatively high among MSM adults, no studies have evaluated patterns of substance use among MSM adults in Mexico……exhibit relatively distinct patterns of substance use), it seems to be more fitful after the second and third paragraphs or all the way to the end of the introduction.
4. For substance use, authors mentioned cigarette use, how about other tobacco products (e.g., e-cigs, cigars, etc.)? If the data has that information, then it should be reflected in the methods.
5. Suggest merge paragraphs two and three, add another paragraph to talk about the literature of risk behaviors related to this study. There is lack of rationale for the outcomes in the study aims.
6. For methods, this study restricted to 1850 MSM (same as the number shown in the recruitment). Is there any missing value in the analysis and how did you deal with that?
7. Please include more details of the three classes as well as the steps/procedures in analysis plan. In addition, please describe the significance level, how to report the statistics (e.g., OR).
8. For discussion, I believe reporting “being HIV positive” results will be more interesting. The description of three classes could be moved to methods part.
9. “We found no association of CAS sexual behaviors and serosorting between the 258 Class 1 and 2.” Should authors make a little bit explanation?
10.Why this study is important? How will the study findings be applied for future research and police making? Please indicate more details in implications.
Author Response

(The authors gave the same response as above.)

Round 2
Reviewer 2 Report
The authors have addressed this reviewer's comments and concerns and provided clarification on issues that needed clarification.
Reviewer 3 Report
This manuscript has been greatly improved. I think it is ready to be accepted.